# Physical Activity after Colorectal Cancer Diagnosis and Mortality in a Nationwide Retrospective Cohort Study

**DOI:** 10.3390/cancers13194804

**Published:** 2021-09-25

**Authors:** Meesun Lee, Yunseo Lee, Doeun Jang, Aesun Shin

**Affiliations:** 1Department of Medicine, Seoul National University College of Medicine, Seoul 03080, Korea; gdnoon@snu.ac.kr (M.L.); temari7@naver.com (Y.L.); 2Department of Preventive Medicine, Seoul National University College of Medicine, Seoul 03080, Korea; doeunj@snu.ac.kr; 3Cancer Research Institute, Seoul National University, Seoul 03080, Korea; 4Integrated Major in Innovative Medical Science, Graduate School of Seoul National University, Seoul 03080, Korea

**Keywords:** colorectal neoplasms, cancer, physical activity, exercise, survival

## Abstract

**Simple Summary:**

Physical activity can help to prevent colorectal cancer, but its importance after cancer diagnosis has not been validated. In this nationwide insurance data-based study of 43,596 colorectal cancer patients, a high level of physical activity after colorectal cancer diagnosis was negatively associated with a risk of death in both colon and rectal cancer patients, particularly in the surgically treated group. Our findings support the importance of the physical activity among colorectal cancer patients.

**Abstract:**

Physical activity reduces the risk of colon cancer, but its prognostic impact after cancer diagnosis remains unclear. To evaluate the association between post-diagnosis activity and cause-specific mortality, we reconstructed a colorectal cancer patient cohort from the 2009–16 Korean National Health Insurance Service (NHIS) database. Subgroup analyses were performed by treatment group. In total, 27,143 colon cancer patients and 16,453 rectal cancer patients were included in the analysis (mean follow-up, 4.3 years; median 4.0 years). In the surgically treated group, a high level of activity (the weighted sum of the frequencies for walking, moderate, and vigorous activity greater than or equal to 3 times/week) was inversely associated with all-cause mortality (colon cancer: HR, 0.79; 95% CI, 0.72 to 0.88; rectal cancer: HR, 0.75; 95% CI, 0.66 to 0.86) and colorectal cancer-specific mortality (colon cancer: HR, 0.85; 95% CI, 0.76 to 0.97; rectal cancer: HR, 0.77; 95% CI, 0.66 to 0.90). No significant results were shown for cardiovascular disease-specific mortality. No association was shown in patients who received chemoradiotherapy without surgery. The present study may provide evidence for post-diagnosis physical activity as a prognostic factor in colorectal cancer, particularly in surgically treated early-stage patients.

## 1. Introduction

Colorectal cancer is the one of the most common cancer types, with an annual incidence of approximately 185 million cases worldwide [1]. In South Korea, it is the second-most common cancer in terms of incidence, with a relative 5 year survival rate of 75.9% [2]. 

Among many lifestyle factors, physical activity is a preventive factor for colon cancer [3,4,5,6]. Increasing evidence has also shown the role of physical activity in colorectal cancer prognosis. Inverse associations between physical activity and mortality have been suggested across different types of activity, from exercise to leisure time or recreational physical activity, and tumor characteristics including site, stage, and molecular profiles [7,8,9,10,11,12,13,14,15,16,17,18]. However, large-scale population-based studies comprehensively and conjointly considering potential variation by cancer treatment and lifestyle variables are scarce, particularly regarding the East Asian population where the prevalence of insufficient physical activity is higher than the global average [19]. 

This study assessed the association between post-diagnosis physical activity and overall and cause-specific mortality in a large retrospective cohort of adult colorectal cancer patients between 2009 and 2016 from the Korean National Health Insurance Service (NHIS) database. Additionally, we analyzed the association by causes of death and according to cancer treatment groups to identify patients who could best benefit from physical activity.

## 2. Materials and Methods

### 2.1. Database and Patient Selection

This study was retrospectively conducted based on the NHIS database. The NHIS covers 98% of the Korean population, and its research database includes information on medical resource utilization, demographic variables, and national health examinations [20]. 

Colorectal cancer patients were defined as subjects with records for hospitalization with diagnostic codes corresponding to colorectal cancer (ICD-10 code C18–20) as well as treatment such as surgery, radiotherapy, or chemotherapy [21]. Patients with an initial cancer claim from 1 January 2009 to 31 December 2016 were included in the study. The date of cancer diagnosis was defined as the date of the first claim with hospitalization for colorectal cancer. We excluded patients without information on activity before or after diagnosis, with missing life-style variables, who were diagnosed before the age of 20 years, without follow-up, or with concurrent diagnoses of colon and rectal cancers. For cancer site-stratified analysis, the diagnosis with ICD-10 code C18 was classified as colon cancer, and C19–C20 was classified as rectal cancer. 

### 2.2. Physical Activity Measurement

Physical activity was measured during the biennial National Health Screening, which included questions on the weekly frequency of vigorous, moderate intensity physical activity and walking performed over a minimal duration of 20 to 30 min. Considering the relative energy expenditure of different activity types, the score for moderate exercise and two times the score for vigorous exercise were added to generate an activity score for individuals aged younger than 65 years. For individuals aged 65 years and older, the scores for moderate exercise and walking were added to two times the score for vigorous exercise. Individuals with a total score greater than or equal to 3 were classified as “high level” exercisers, while the rest were classified as “low level” exercisers. This score was used to reflect the global activity recommendation by the World Health Organization (WHO).

The post-diagnosis physical activity level was extracted from the questionnaire from the first health checkup after cancer diagnosis. Additionally, the pre-diagnosis physical activity was obtained from the closest health checkup before cancer diagnosis. 

### 2.3. Mortality Follow-Up

For each patient, follow-up started at the time of post-diagnosis activity measurement and ended at death or the end of the cohort study (31 December 2016). We linked the patient data with death certificates from the Korean National Statistical Office, from which the dates and the specific causes of death were extracted. All-cause mortality was defined as any death with a recorded date. Colorectal cancer-related mortality and cardiovascular disease-related mortality were identified with ICD-10 codes of the cause of death, C18–C20 for colorectal cancer, and I20-I25 (coronary heart disease) and I60–I69 (stroke) for cardiovascular diseases. 

### 2.4. Covariates

Baseline characteristics were assessed until the post-diagnosis health examination, when physical activity was measured and follow-up period started. Age, sex, and insurance premium level (in tertiles, including Medicare recipients in the first tertile) were provided from the insurance claim and the eligibility data at the time of cancer diagnosis. The body mass index (BMI) was calculated from the height and the weight measured during the first post-diagnosis checkup. BMI was categorized as <18.5 kg/m^2^, 18.5–22.9 kg/m^2^, 23.0–24.9 kg/m^2^, and ≥25.0 kg/m^2^ based on the WHO Asia-Pacific obesity classification [22]. Information on cigarette smoking and alcohol consumption was collected using self-administered questionnaires during the health checkup. Alcohol consumption data were converted into the number of drinks consumed per day and then classified into three groups (nondrinker/social/heavy) in accordance with the U.S. alcohol and alcoholism guidelines [23]. Men were considered to drink heavily if they consumed at least 2 drinks per day on average, and women were considered to drink heavily if they consumed at least 1 drink per day on average. Otherwise, the subjects were considered to drink socially. 

The Charlson Comorbidity Index (CCI) was calculated from claims data of at least one hospitalization for specified comorbidities, including cancer, during the year before colorectal cancer diagnosis [24]. 

Information concerning the cancer stage, which is the most important prognostic factor, was not available from the NHIS database. Instead, we sought to address this issue by stratifying the patients according to the type of cancer treatment they received. The treatment groups were classified into three types: surgery only, surgery with chemotherapy and/or radiotherapy, and chemotherapy with or without radiotherapy.

### 2.5. Statistical Analyses

The distribution of general characteristics was described as *n* (%) for colon and rectal cancers separately. Chi-square test was used to compare the characteristics of the activity groups with each cancer. Using the low activity group as a reference, Cox proportional hazards models were used to estimate the HRs and the 95% CIs of all-cause, colorectal cancer-related, and cardiovascular disease-related mortality. Proportional hazard assumptions were confirmed using log minus log survival plots as well as the Schoenfeld residual method [25]. Stratified analysis was performed according to the treatment group. 

The hazards models were adjusted for patient age, sex, BMI, smoking status, level of alcohol consumption, insurance premium, CCI, physical activity before cancer diagnosis, and treatment group. The potential confounding effects of each covariate were tested using Chi-square tests and univariate survival analyses.

For sensitivity analysis, an alternative measurement of physical activity was tested by accounting for the sum of frequencies for moderate-to-vigorous activity instead of the weighted sum.

To exclude the possibility of reverse causality between post-diagnosis physical activity and the health status of the patient, an additional analysis was performed by excluding patients with less than a half-year of follow-up. 

A two-sided *p*-value of <0.05 was considered statistically significant. All the analyses were performed using SAS version 9.4 and SAS Enterprise Guide (SAS Institute Inc., Cary, NC, USA).

## 3. Results

### 3.1. Baseline Characteristics by Physical Activity Category 

In total, 27,143 colon cancer patients and 16,453 rectal cancer patients were identified from the NHIS database after applying the exclusion criteria (Figure 1). A comparison of the demographic and the clinical characteristics of patients before and after selection is shown in Table 1. Briefly, the final study population was younger and more likely to be a male, to be in the lower insurance premium tertiles, or to receive surgery only. 

Their baseline characteristics were assessed from the closest health checkup before diagnosis until the first health checkup after diagnosis. The mean age at the time of cancer diagnosis was 62.7 years (standard deviation, 10.4 years). The patients were followed for a mean duration of 4.3 years (median, 4.0 years). The physical activity level was assessed at an average of 1.8 years after cancer diagnosis.

The baseline characteristics of the colorectal cancer patients are shown according to the cancer location and the physical activity categories (Table 2). In both colon and rectal cancers, physically active patients were significantly older, had a greater male proportion, were less underweight or obese, had a smaller ratio of current smokers and heavy drinkers, had greater income levels, had a higher pre-diagnosis activity level, and included a lower proportion of patients treated with chemotherapy with or without radiotherapy. The distribution of the CCI was not significantly different. For both cancers, the most common treatment was surgery without additional radiotherapy or chemotherapy, which was performed on 74.5% of colon cancer patients and 60.7% of rectal cancer patients. Fewer than 2% of the patients had no record of surgical resection but received chemotherapy with or without radiotherapy. 

### 3.2. Physical Activity and All-Cause, Colorectal Cancer, and Cardiovascular Mortality

During 106,927 person-years of follow up, 1673 deaths by colon cancer patients and 971 deaths by rectal cancer patients were observed. Among them, 69.6% of the deaths were due to colorectal cancer, and 3.0% were due to other causes including cardiovascular diseases such as coronary heart diseases (International Classification of Diseases, Tenth Revision (ICD-10) code I20–I25) or stroke (ICD-10 code I60–I69). After adjustment, both colon and rectal cancer patients with a high level of activity showed lower all-cause mortality (colon cancer: hazard ratio (HR), 0.79; 95% confidence interval (95% CI), 0.72 to 0.88; rectal cancer: HR, 0.75; 95% CI, 0.66 to 0.86) (Table 2; Figure 2). A high level of activity was similarly associated with a lowered risk of colorectal cancer-specific mortality (colon cancer: HR, 0.85; 95% CI, 0.76 to 0.97; rectal cancer: HR, 0.77; 95% CI, 0.66 to 0.90) (Table 3). However, no significant association was found between cardiovascular disease-related deaths and physical activity (colon cancer: HR, 0.87; 95% CI, 0.49 to 1.54; rectal cancer: HR, 0.88, 95% CI, 0.40 to 1.94) (Table 3). Additionally, sensitivity analysis for physical activity was performed by considering only the sum of the frequency of moderate-to-vigorous activity. Again, all-cause mortality and colorectal cancer-related mortality were lower in the physically active patients but not cardiovascular disease-related mortality (Appendix A).

The difference in mortality could also be due to reverse causation such as premorbid or elderly patients only having a limited amount of physical capacity. This possibility was explored by excluding patients who died within 6 months of follow-up. Even after excluding them, the association between physical activity and overall mortality was still significant (colon cancer: HR, 0.82; 95% CI, 0.73 to 0.91; rectal cancer: HR, 0.79; 95% CI, 0.69 to 0.91). The association was also consistent in patients aged 65 years or older (colon cancer: HR, 0.76; 95% CI, 0.67 to 0.86; rectal cancer: HR, 0.76; 95% CI, 0.64 to 0.90).

Analysis of the effect of pre-diagnostic physical activity on colorectal cancer prognosis revealed that satisfying the activity recommendation during the pre-diagnosis period was associated with significantly lower all-cause, colorectal cancer-related, and cardiovascular disease-related mortality among colon cancer patients. In rectal cancer, however, the association was only significant for all-cause mortality (Appendix A). The results of the analyses for the change between pre- and post-diagnostic physical activity were conflicting, but a decrease in physical activity after cancer diagnosis was consistently associated with higher mortality in both colon and rectal cancer patients (Appendix A).

### 3.3. Physical Activity and Colorectal Cancer Treatment

In the stratified analysis by cancer treatment, active patients who only received surgical resection had an adjusted hazard ratio of 0.75 (95% CI, 0.65 to 0.87), and those who received surgery as well as chemotherapy and/or radiotherapy had an adjusted hazard ratio of 0.84 with a high level of activity (95% CI, 0.73 to 0.97) for colon cancer. However, physical activity was not significantly associated with mortality in those who did not receive surgery but received chemotherapy with or without radiotherapy. Likewise, in rectal cancer patients, physical activity was associated with lower mortality in surgically treated patients but not in the nonsurgically treated patients (Table 3; Figure 2).

## 4. Discussion

Our analyses uncovered an inverse association between post-diagnosis physical activity and mortality in both colon and rectal cancer patients. This finding implies that physical activity, apart from being an independent protective factor against colorectal cancer, may also serve as a prognostic determinant of colorectal cancer. Additionally, our results suggest that activity may be most beneficial in surgically treatable patients, who are presumably early-stage cancer patients with longer survival times after cancer treatment.

The benefits of physical activity may have resulted from changing the biological processes related to cancer progression. Physical activity was suggested to affect tumor metabolism in the acute phase and then induce immunological control of tumor growth in the long term [26]. The Akt/mTOR signaling pathway, a mechanism potentially related to colorectal cancer growth and metastasis, was shown to be differentially regulated during exercise [27,28]. Inflammatory markers such as TNF-α and IL-6 decreased after exercise training in a study of breast cancer survivors [29]. A mouse model study also suggested that natural killer cells were mobilized during exercise [30]. The accumulation of these effects could have led to better survival of the active patients. 

Additionally, physical activity may be crucial in the quality of life of patients. Active cancer patients are generally shown to experience better cardiovascular recovery during treatment [31]. They also have less cancer-related fatigue and fewer side effects such as treatment-induced anemia [32,33]. In accordance with these results, the amount of anxiety or depression is lower in patients with higher amounts of physical activity [34]. Combined with our results, these studies provide evidence that physical activity may be crucial for both the quantity and the quality of life. However, this interpretation requires further evaluation, because previous studies primarily focused on the effects of exercise, which only included voluntary and regular body movements and not all types of physical activity.

The similarity of results for colon and rectal cancer patients should be noted, because the role of physical activity in cancer prevention is established for only colon cancer [35]. These cancers differ not only in their anatomical origin but also in several important clinical features. The role of radiotherapy is limited in colon cancer treatment, but the effect of adjuvant systemic treatment on survival is more evident against this cancer. The most common metastatic site of colorectal cancers is the liver, but rectal cancer may also initially metastasize to the lungs because of its drainage into the inferior vena cava rather than the portal venous system [36]. Our study suggests evidence that, despite these differences, physical activity acts as a common prognostic factor in these cancers.

In our study, patients who did not receive surgery received no benefit from high levels of activity. An explanation for this result is that the stages of cancer in this group were too far advanced to be moderated by physical activity. Another possibility is that these patients had more comorbidities that overwhelmed the benefits of physical activity. However, the results remained unchanged even after adjusting for the CCI. Finally, surveillance or treatment for other primary cancers, recorded in 10% of colorectal cancer patients, could have added heterogeneity to the data. When we omitted patients with at least one record of other cancers (identified by the ICD-10 “C” code) within three years before the colorectal cancer diagnosis, the association between activity and mortality in nonsurgically treated patients was significant for colon cancer but not rectal cancer (colon cancer: HR, 0.72; 95% CI, 0.42 to 0.98; rectal cancer: HR, 0.75; 95% CI, 0.45 to 1.17). Careful interpretation is required because of the small number of patients treated without surgery [37]. However, although other studies suggested a significant reduction in mortality in advanced but active cancer patients [38,39,40], our results suggest that exercise rehabilitation may be most beneficial during earlier stages of cancer in which medical treatments such as surgery, chemotherapy, or radiotherapy are all available. 

Despite the decrease in all-cause mortality as well as colorectal cancer-related mortality, we did not observe a decrease in cardiovascular disease-related mortality, in contrast to results from the previous literature [41]. The reason may be due to the insufficiency of follow-up to capture the slow development of cardiovascular events, because we observed 3% of deaths from cardiovascular diseases, whereas 10% to 20% of deaths were reported from other study of cancer patients [42]. Another possibility is that the noncancer death profiles are truly unaffected by activity in the cancer patients. Additional studies with longer follow-up could provide further insight into how the causes of death in colorectal cancer patients change according to activity level. 

One of the strengths of this study relates to the large size of the cohort. A meta-analysis on the relationship between post-diagnosis physical activity and colorectal cancer showed that the number of total subjects in previous studies was in the hundreds to thousands [43]. By contrast, our results clarified the relationship between activity and mortality in a larger cohort that was representative of the national population and included more than 40,000 patients. Furthermore, the use of Korean NHIS data allowed us to evaluate the effect of physical activity on the Asian population, a phenomenon that was not sufficiently studied previously.

Additionally, the large size of the cohort enabled us to perform stratified analyses on cancer sites and treatment options without decreasing the reliability of the statistical results. Cancer site-stratification was used to distinguish among the results from colon and rectal cancers. The treatment options were used as indicators of the cancer stage, the primary prognostic factor of colorectal cancer that was not directly available from our data. These stratifications made it possible to demonstrate the adjuvant effect of physical activity in detail, the finding of which could be meaningful during clinical applications of the data. 

There are several limitations that must be considered. First, selection bias may occur during study subject selection because many colorectal patients were excluded because of the absent or the incomplete post-diagnosis health checkups. The largest difference was shown in the distribution of treatment options, likely because of the short survival in patients with advanced cancer (Table 1). Although no differences were found in the proportion of the social/heavy alcohol drinkers between total and final study populations, the final study population showed a slightly higher proportion of former smokers and high pre- and post-diagnostic PA levels. The study population may have a more desirable lifestyle than the excluded patients, and it is unlikely that the internal validity for comparing PA groups was affected. 

Second, the retrospective design of the study made it challenging to incorporate residual confounding factors such as performance status and validate the causality between mortality and different factors that were included in the analyses. The level of physical activity may only reflect health status but may not be a determinant of health status. To counter this interpretation, we adjusted our model for diverse prognostic factors of colorectal cancer and performed sensitivity analysis to confirm our results. An analysis with residual confounding factors and time-varying covariates can further strengthen the data. Third, our data only included information on the minimal duration and intensity of physical activity. This approach precluded the evaluation of activities in the standard measurement of MET minutes per week or the analysis on the dose effect of the activity. When we validated our activity criteria by comparing the achievement rates using the national cancer statistics, the proportion of active patients was larger in our study than the national average of 57.1%, suggesting that our criteria might have been more relaxed [44]. However, our finding could also be explained by a greater interest in activity among colorectal cancer patients, and analyses using an alternative definition of activity showed a similar relationship to survival. Fourth, a relatively long interval existed between cancer diagnosis and health checkup (average, 1.8 years; standard deviation, 1.2 years), possibly leading to a bias toward patients with relatively better survival sufficient to attend health checkups. Finally, the bias due to competing risk between the causes of death cannot be completely excluded. 

## 5. Conclusions

A high level of post-diagnosis physical activity was associated with lower mortality in surgically treated colorectal patients, considering the variation by cancer treatment and lifestyle factors. Future research is necessary to consider the interaction among activity, performance status, and cancer stage and to understand the underlying molecular mechanisms supporting the clinical role of post-diagnosis activity.

## Figures and Tables

**Figure 1 cancers-13-04804-f001:**
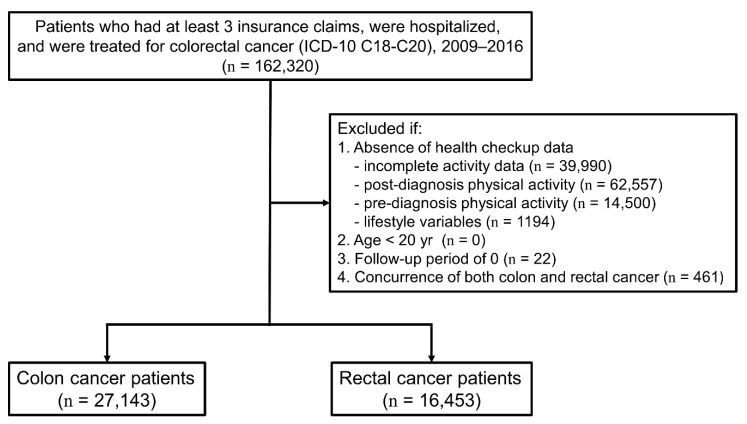
Flow chart to selecting study subjects from the Korean National Health Insurance Service Database. ICD-10 indicates International Classification of Diseases, Tenth Revision.

**Figure 2 cancers-13-04804-f002:**
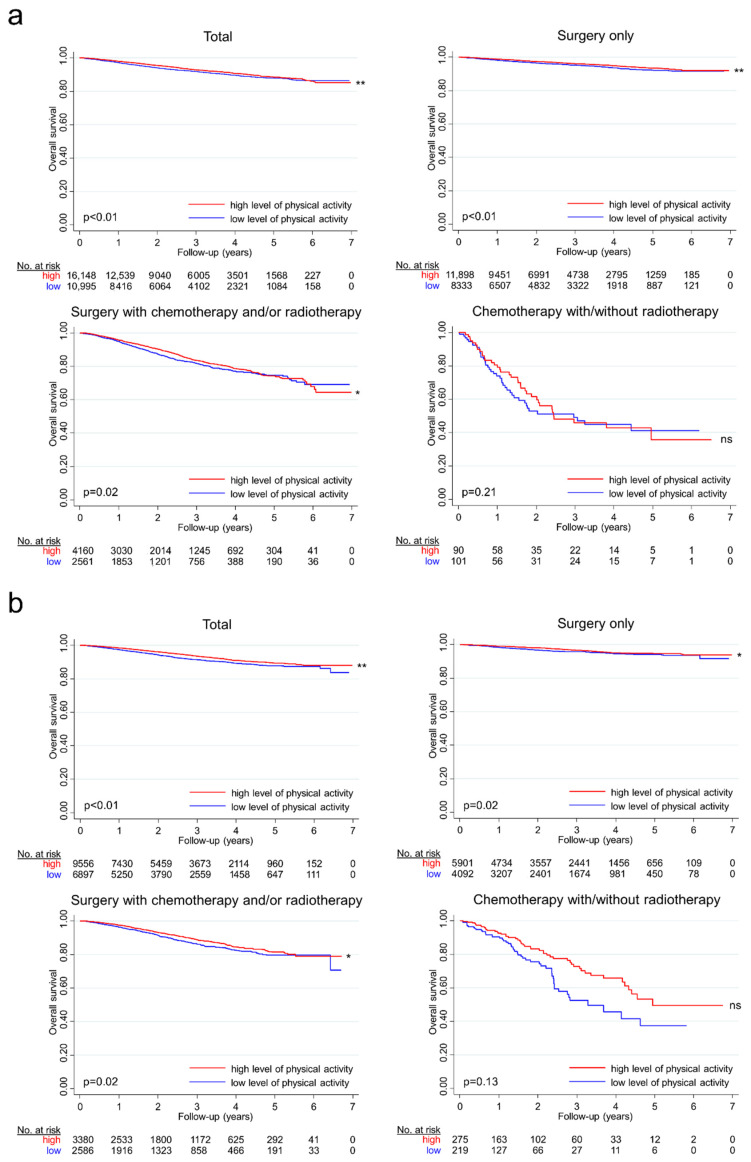
Kaplan–Meier survival curves as stratified by physical activity groups for colon and rectal cancer. Kaplan–Meier survival curves for colon cancer patients (**a**) and rectal cancer patients (**b**), stratified by the following treatment groups: total, surgery only, surgery with chemotherapy and/or radiotherapy, and chemotherapy with or without radiotherapy. The red line represents the survival probability of patients with a high level of activity (the weighted sum of the frequencies for walking, moderate, and vigorous activity greater than or equal to three times/week), and the blue line represents the survival probability of patients with a low level of activity. **, *, and ns next to graphs indicate *p*-values lower than 0.01, *p*-values lower than 0.05, and nonsignificant results, respectively. The results indicate that satisfying the recommended level of activity is associated with a lowered risk of mortality in colorectal cancer patients, particularly in patients who received surgery with or without other treatment options such as chemotherapy or radiotherapy.

**Table 1 cancers-13-04804-t001:** Comparison of the demographic and the clinical characteristics of colorectal cancer patients before and after selection, 2009–2016.

Characteristics	Before Selection ^a^		After Selection ^b^		*p*
	(*n* = 162,320)		(*n* = 43,596)		
	*n*	%	*n*	%	
Age, ≥60 years	107,249	66.21	27,426	62.92	<0.01
missing value ^c^	280	(0.17)	0	(0)	
Sex					<0.01
male	97,190	59.99	27,077	62.11	
female	64,818	40.01	16,519	37.89	
missing value	312	(0.19)	0	(0)	
BMI, kg/m^2^					0.01
<18.5	2020	3.37	1366	3.13	
18.5–22.9	21,623	36.12	15,455	35.45	
23.0–24.9	15,616	26.09	11,504	26.39	
≥25.0	20,601	34.42	15,271	35.03	
missing value	102,460	(63.12)	0	(0)	
Smoking					<0.01
never	35,855	59.88	26,135	59.95	
former	18,074	30.19	13,485	30.93	
current	5946	9.93	3976	9.12	
missing value	102,445	(63.11)	0	(0)	
Alcohol consumption					0.29
nondrinker	48,583	81.25	35,507	81.45	
social	7308	12.22	5351	12.27	
heavy	3901	6.53	2738	6.28	
missing value	102,528	(63.16)	0	(0)	
Insurance premium					<0.01
1st tertile	34,653	22.95	11,080	25.41	
2nd tertile	43,254	28.65	12,232	28.06	
3rd tertile	73,063	48.40	20,284	46.53	
missing value	11,350	(6.99)	0	(0)	
Pre-diagnosis PA					0.39
low	51,654	47.96	20,804	47.72	
high	56,043	52.04	22,792	52.28	
missing value	54,623	(33.65)	0	(0)	
Post-diagnosis PA					<0.01
low	25,376	42.46	17,892	41.04	
high	34,383	57.54	25,704	58.96	
missing value	102,561	(63.18)	0	(0)	
Treatment					<0.01
surgery only	89,570	57.30	30,224	69.33	
surgery with chemotherapy and/or radiotherapy	56,245	35.98	12,687	29.10	
chemotherapy with or without radiotherapy	10,494	6.72	685	1.57	
missing value	6011	(3.70)	0	(0)	

Abbreviations: PA, physical activity; *n*, number of patients; BMI, body mass index. ^a^ Colorectal cancer patients who were collected as the initial study subjects from the Korean National Health Insurance Service Database before applying the exclusion criteria (absence of health checkup data, under the age of 20 years, follow-up period of 0, concurrence of both colon and rectal cancer). ^b^ Colorectal cancer patients who were selected as the final study subjects after applying the exclusion criteria. ^c^ The percentage in parentheses is the fraction of missing values for each variable.

**Table 2 cancers-13-04804-t002:** Demographic and clinical characteristics of colorectal cancer patients by post-diagnosis physical activity levels, 2009–2016.

Characteristics	Colon Cancer					Rectal Cancer				
	low PA^a^		high PA ^b^		*p*	low PA		high PA		*p*
	(*n* = 10,995)		(*n* = 16,148)			(*n* = 6897)		(*n* = 9556)		
	*n*	%	*n*	%		*n*	%	*n*	%	
Age, years					<0.01					<0.01
20–29	17	0.2	24	0.2		14	0.2	10	0.1	
30–39	171	1.6	238	1.5		111	1.6	180	1.9	
40–49	914	8.3	1259	7.8		636	9.2	928	9.7	
50–59	3265	29.7	3660	22.7		2286	33.1	2457	25.7	
60–69	3724	33.9	5531	34.3		2297	33.3	3331	34.9	
70–79	2405	21.9	4846	30.0		1317	19.1	2404	25.2	
80–89	488	4.4	579	3.6		230	3.3	240	2.5	
90–99	11	0.1	11	0.1		6	0.1	6	0.1	
Sex					<0.01					<0.01
male	5912	53.8	10,529	65.2		4044	58.6	6592	69.0	
female	5083	46.2	5619	34.8		2853	41.4	2964	31.0	
BMI, kg/m^2^					<0.01					<0.01
<18.5	371	3.4	406	2.5		266	3.9	323	3.4	
18.5–22.9	3615	32.9	5503	34.1		2599	37.7	3738	39.1	
23.0–24.9	2783	25.3	4481	27.8		1716	24.9	2524	26.4	
≥25.0	4226	38.4	5758	35.7		2316	33.6	2971	31.1	
Smoking					<0.01					<0.01
never	7225	65.7	9437	58.4		4288	62.2	5185	54.3	
former	2632	24.0	5482	34.0		1782	25.8	3589	37.6	
current	1138	10.4	1229	7.6		827	12.0	782	8.2	
Alcohol consumption					<0.01					<0.01
nondrinker	9047	82.3	12,990	80.4		5709	82.8	7761	81.2	
social	1178	10.7	2185	13.5		728	10.6	1260	13.2	
heavy	770	7.0	973	6.0		460	6.7	535	5.6	
Insurance premium					<0.01					<0.01
1st tertile	2929	26.6	3905	24.2		1909	27.7	2337	24.5	
2nd tertile	3279	29.8	4178	25.9		2130	30.9	2645	27.7	
3rd tertile	4787	43.5	8065	49.9		2858	41.4	4574	47.9	
CCI					0.30					0.55
0	8504	77.3	12,550	77.7		5424	78.6	7526	78.8	
1	1369	12.5	2037	12.6		828	12.0	1179	12.3	
2	430	3.9	562	3.5		231	3.4	285	3.0	
≥3	692	6.3	999	6.2		414	6.0	566	5.9	
Pre-diagnosis PA					<0.01					<0.01
low	6836	62.2	5919	36.7		4319	62.6	3730	39.0	
high	4159	37.8	10,229	63.4		2578	37.4	5826	61.0	
Treatment					<0.01					<0.01
surgery only	8333	75.8	11,898	73.7		4092	59.3	5901	61.8	
surgery with chemotherapy and/or radiotherapy	2561	23.3	4160	25.8		2586	37.5	3380	35.4	
chemotherapy with or without radiotherapy	101	0.9	90	0.6		219	3.2	275	2.9	

Abbreviations: PA, physical activity; *n*, number of patients; BMI, body mass index; CCI, Charlson comorbidity index. ^a^ Patients with a low level of activity (the weighted sum of the frequencies for walking, moderate, and vigorous activity fewer than three times/week). ^b^ Patients with a high level of activity (the weighted sum of the frequencies for walking, moderate, and vigorous activity greater than or equal to three times/week).

**Table 3 cancers-13-04804-t003:** Post-diagnosis physical activity and all-cause/colorectal cancer/cardiovascular mortality in colon and rectal cancer patients.

Treatment	PA	*n*	Person-Years	All-Cause			Colorectal Cancer			Cardiovascular		
				Events	HR	(95% CI)	Events	HR	(95% CI)	Events	HR	(95% CI)
Colon cancer												
Total		27,143	66,458.54	1673			1155			51		
	Low ^a^	10,995	26,780.26	734	1 (Ref)		488	1 (Ref)		23	1 (Ref)	
	High ^b^	16,148	39,678.27	939	0.79	(0.72, 0.88)	667	0.85	(0.76, 0.97)	28	0.87	(0.49, 1.54)
Surgery only		20,231	51,613.34	754			380			44		
	low	8333	21,148.54	352	1 (Ref)		172	1 (Ref)		20	1 (Ref)	
	high	11,898	30,464.80	402	0.75	(0.65, 0.87)	208	0.81	(0.66, 1.00)	24	0.86	(0.46, 1.59)
Surgery with chemotherapy and/or radiotherapy		6721	14,495.30	841			703			7		
	low	2561	5454.82	341	1 (Ref)		278	1 (Ref)		3	1 (Ref)	
	high	4160	9040.48	500	0.84	(0.73, 0.97)	425	0.89	(0.76, 1.04)	4	0.92	(0.20, 4.28)
Chemotherapy with or without radiotherapy		191	349.90	78			72			0		
	low	101	176.91	41	1 (Ref)		38	1 (Ref)		0		
	high	90	172.99	37	0.74	(0.46, 1.19)	34	0.70	(0.43, 1.16)	0		
Rectal cancer												
Total		16,453	40,468.54	971			686			27		
	low	6897	16,730.29	463	1 (Ref)		328	1 (Ref)		11	1 (Ref)	
	high	9556	23,738.25	508	0.75	(0.66, 0.86)	358	0.77	(0.66, 0.90)	16	0.88	(0.40, 1.94)
Surgery only		9993	26,008.18	318			148			19		
	low	4092	10,566.05	149	1 (Ref)		74	1 (Ref)		7	1 (Ref)	
	high	5901	15,442.12	169	0.75	(0.60, 0.95)	74	0.69	(0.49, 0.96)	12	1.00	(0.38, 2.63)
Surgery with chemotherapy and/or radiotherapy		5966	13,666.00	552			447			8		
	low	2586	5845.20	262	1 (Ref)		207	1 (Ref)		4	1 (Ref)	
	high	3380	7820.80	290	0.81	(0.68, 0.96)	240	0.86	(0.71, 1.04)	4	0.73	(0.17, 3.05)
Chemotherapy with or without radiotherapy		494	794.36	101			91			0		
	low	219	319.03	52	1 (Ref)		47	1 (Ref)		0		
	high	275	475.33	49	0.72	(0.46, 1.11)	44	0.71	(0.45, 1.12)	0		

Adjusted for age at diagnosis, sex, BMI, smoking, alcohol consumption, insurance premium, Charlson comorbidity index, pre-diagnosis physical activity. For all patients, treatment options were also adjusted. Abbreviations: PA, physical activity; *n*, number of patients; HR, hazard ratio; 95% CI, 95% confidence interval; Ref, reference. ^a^ Patients with a low level of activity (the weighted sum of the frequencies for walking, moderate, and vigorous activity fewer than three times/week). ^b^ Patients with a high level of activity (the weighted sum of the frequencies for walking, moderate, and vigorous activity greater than or equal to three times/week).

## Data Availability

To protect potentially identifiable information on subjects, ethical approval is required to access data. The data are available from the Korean NHIS database for researchers who meet the criteria for access.

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
