# Peer review of "Physical Activity after Colorectal Cancer Diagnosis and Mortality in a Nationwide Retrospective Cohort Study"

_cancers, 2021, doi:10.3390/cancers13194804_

Round 1
Reviewer 1 Report
Physical Activity After Colorectal Cancer Diagnosis and Mortality in a Nationwide Retrospective Cohort Study
Meesun Lee et al aimed to assess the association between post-diagnosis physical activity and mortality in a large retrospective cohort of colorectal cancer patients from the Korean National Health Insurance Service (NHIS) database. In addition, we analyzed the association by causes of death and cancer treatment groups to identify patients who could best benefit from physical activity.
This is an interesting paper which addresses an hot topic in modern medicine such as the “therapeutic use” of physical activity. However the paper has several flags that weaken its validity.
Major issues
- The objective does not state the association of the exposure (post-diagnosis physical activity) with cause-specific mortality. Moreover, an epidemiologic study is always a measurement exercise anìd needs statement about person, time and place. In this case t”he study was aimed to estimate the association between PA and end-points in person who were 20 yo or older from XX to XX in Korea.”
- This is a typical competing risk scenario which is nor described in the statistical analysis section neither in the analysis but is considered in the results and discussion section. Then, the estimators obtained setting say a cardiovascular end-point and the other causes as censored produced biased estimators. Conceptually are different entities and, if not addressed conclusions could be wrong. It is necessary at least to discuss and include them as limitations.
- Selection bias must be more intensively discussed. There are thousands of missing values for variables which are risk factors such as alcohol use. Moreover, it is necessary to consider that participants might have given socially desirable answers for life-style variables.
- Authors must considered a multiple imputation approach to overcome this issue.
- There is no mention about methods to test proportional hazard assumption. Other than the graphical visual inspection by log minus log survival plots it would be interesting to use Schoenfeld residuals to test each variable and global test for this assumption as well as log-rank tests for each variable considered. Moreover, no log minus log survival plot has been included in the paper.
- Caution must be paid when the term confounder/confounding is used as this is a precise concept in epidemiology and should be define a priori to control it in the study design or in the analytical step.
- Define the measurement of follow-up. There is only one mention of duration of follow-up which is reported as a mean (without a SE). The most convenient way to measure duration of follow-up is to use a median as the time-to-event is usually asymmetrical.
- Authors mention in the the methods section (2.4 Covariates) “Information about the cancer stage, which is the most important prognostic factor, was not available from the NHIS database. Instead, we sought to address this issue by stratifying the patients according to the type of cancer treatment they received. The treatment groups were classified into three types: surgery only, surgery with chemotherapy and/or radiotherapy, and chemotherapy with or without radiotherapy.” In the discussion they mention “Second, retrospective design of the study made it difficult to incorporate residual confounding such as performance status or stage, and validate the causality between mortality and different factors that were included in the analyses.” If cancer therapy has been considered as proxy-variable to measure stage this paragraph is redundant. Please rephrase.
- This is a prospective study non a retrospective one. The exposure has been assessed at the beginning of follow-up and patients were followed-up to the end-point (deaths) or the final of the study whichever occurred first. Then, this is a non-concurrent prospective follow-up study.
- There are several paragraphs which have no sense. Please make an editing by an English native speaker is needed.
Minor issues
- There are several typos along the paper. The most frequent one being words separated by a dash.
I hope authors can overcome these issues to produce a very high quality paper.
Author Response
Major issues
The objective does not state the association of the exposure (post-diagnosis physical activity) with cause-specific mortality. Moreover, an epidemiologic study is always a measurement exercise anìd needs statement about person, time and place. In this case t”he study was aimed to estimate the association between PA and end-points in person who were 20 yo or older from XX to XX in Korea.”
Authors’ response: To accommodate the reviewer’s suggestion, we modified the sentence in the Introduction section as follows:
This study assessed the association between postdiagnosis physical activity and overall and cause-specific mortality in a large retrospective cohort of adult colorectal cancer patients between 2009 and 2016 from the Korean National Health Insurance Service (NHIS) database.
This is a typical competing risk scenario which is nor described in the statistical analysis section neither in the analysis but is considered in the results and discussion section. Then, the estimators obtained setting say a cardiovascular end-point and the other causes as censored produced biased estimators. Conceptually are different entities and, if not addressed conclusions could be wrong. It is necessary at least to discuss and include them as limitations.
Authors’ response: According to the reviewer’s comment, the following sentences were added to the Discussion section:
Finally, bias due to competing risk between the causes of death cannot be completely excluded.
Selection bias must be more intensively discussed. There are thousands of missing values for variables which are risk factors such as alcohol use. Moreover, it is necessary to consider that participants might have given socially desirable answers for life-style variables.
Authors’ response: According to the reviewer’s comment, the following sentences were added to the Discussion section:
Although no differences were found in the proportion of social/heavy alcohol drinkers between the total and final study populations, the final study population showed a slightly higher proportion of former smokers and high pre- and post-diagnostic PA levels. The study population may have a more desirable lifestyle than the excluded patients, and it is unlikely that the internal validity for comparing PA groups was affected.
Authors must considered a multiple imputation approach to overcome this issue.
Authors’ response: After including colorectal cancer patients with PA information, no subjects with missing information on variables were included in the final model (Table 1). Therefore, we do not believe that multiple imputation is needed.
There is no mention about methods to test proportional hazard assumption. Other than the graphical visual inspection by log minus log survival plots it would be interesting to use Schoenfeld residuals to test each variable and global test for this assumption as well as log-rank tests for each variable considered. Moreover, no log minus log survival plot has been included in the paper.
Authors’ response: The log-log plots were included as supplementary material in the revision, and the sentence in the statistical analysis was revised as follows:
Proportional hazard assumptions were confirmed using log minus log survival plots as well as the Schoenfeld residual method [25].
Caution must be paid when the term confounder/confounding is used as this is a precise concept in epidemiology and should be define a priori to control it in the study design or in the analytical step.
Authors’ response: Potential confounders of the association between physical activity and colorectal cancer survival were selected based on literature review and are described in the statistical analysis section as follows:
The hazards models were adjusted for patient age, sex, BMI, smoking status, level of alcohol consumption, insurance premium, CCI, physical activity before cancer diagnosis, and treatment group. The potential confounding effects of each covariate were tested using chi-squared test and univariate survival analyses.
Define the measurement of follow-up. There is only one mention of duration of follow-up which is reported as a mean (without a SE). The most convenient way to measure duration of follow-up is to use a median as the time-to-event is usually asymmetrical.
Authors’ response: We have added the median follow-up time to the revised manuscript as follows:
Abstract:
In total, 27,143 colon cancer patients and 16,453 rectal cancer patients were included in the analysis (mean follow-up, 4.3 years; median 4.0 years).
Results:
Next, the patients were followed for a mean duration of 4.3 years (median, 4.0 years).
Authors mention in the methods section (2.4 Covariates) “Information about the cancer stage, which is the most important prognostic factor, was not available from the NHIS database. Instead, we sought to address this issue by stratifying the patients according to the type of cancer treatment they received. The treatment groups were classified into three types: surgery only, surgery with chemotherapy and/or radiotherapy, and chemotherapy with or without radiotherapy.” In the discussion they mention “Second, retrospective design of the study made it difficult to incorporate residual confounding such as performance status or stage, and validate the causality between mortality and different factors that were included in the analyses.” If cancer therapy has been considered as proxy-variable to measure stage this paragraph is redundant. Please rephrase.
Authors’ response: The sentence in the Discussion section has been rephrased as follows:
Second, the retrospective design of the study made it challenging to incorporate residual confounding factors such as performance status and validate the causality between mortality and different factors that were included in the analyses.
This is a prospective study non a retrospective one. The exposure has been assessed at the beginning of follow-up and patients were followed-up to the end-point (deaths) or the final of the study whichever occurred first. Then, this is a non-concurrent prospective follow-up study.
Authors’ response: We agree with the reviewer that this study has a prospective nature regarding the exposure-outcome assessment. Because our study was conducted in 2019 by reconstructing the data between 2009 and 2016, we believe it would be better described as a retrospective cohort study.
There are several paragraphs which have no sense. Please make an editing by an English native speaker is needed.
Authors’ response: The original and revised versions were edited by a professional editing service, and the certificate is attached for the reviewer’s reference.
Minor issues
There are several typos along the paper. The most frequent one being words separated by a dash.
I hope authors can overcome these issues to produce a very high quality paper.
Authors’ response: We have now removed the unnecessary dashes that were likely generated during the conversion to the journal format.
Reviewer 2 Report
The Authors should insert and comment the first table showed in the supplementary file to allow readers to evaluate the representativeness of the analyzed sample
Author Response
Authors’ response: According to the reviewer’s comments, the previous Supplementary Table 1 in now Table 1 in the revised manuscript and the corresponding description were added as follows:
Briefly, the final study population was younger and more likely to be male, in the lower insurance premium tertiles or receive surgery only.

Round 2
Reviewer 1 Report
Authors have partially addressed the issues raised by me.
It remains to address confounding and missing values.